# Trends and Interdependence of Solar Radiation and Air Temperature—A Case Study from Germany

Hein Dieter Behr 

Deutscher Wetterdienst, D-63067 Offenbach am Main, Germany; hein-dieter.behr@t-online.de

**Abstract:** This study characterizes the spatiotemporal solar radiation and air temperature patterns and their dependence on the general atmospheric circulation characterized by the North Atlantic Oscillation (NAO) Index in Germany from 1991 to 2015. Germany was selected as the study area because it can be subdivided into three climatologically different regions: the North German lowlands are under the maritime influence of the North and Baltic Seas. Several low mountain ranges dominate Germany's center. In the south, the highest low mountain ranges and the Alps govern solar radiation and air temperature differently. Solar radiation and air temperature patterns were studied in the context of the NAO index using daily values from satellite and ground measurements. The most significant long-term solar radiation increase was observed in spring, mainly due to seasonal changes in cloud cover. Air temperature shows a noticeable increase in spring and autumn. Solar radiation and air temperature were significantly correlated in spring and autumn, with correlation coefficient values up to 0.93. In addition, a significant dependence of solar radiation and air temperature on the NAO index was revealed, with correlation coefficient values greater than 0.66. The results obtained are important not only for studies on the climate of the study area but also for photovoltaic system operators to design their systems. They need to be massively expanded to support Germany's climate neutrality ambitions until 2045.

**Keywords:** solar energy; global warming; decadal variability; climate change mitigation

## 1. Introduction

Changes in atmospheric transmission due to variations in cloudiness and aerosol concentration are the main factors causing trends in incoming solar radiation on a horizontal surface (*SIS*) penetrating the atmosphere and finally being available at the Earth's surface for the generation of solar energy.

Due to aerosol–cloud interactions and aerosol and water vapor's indirect effects on *SIS*, these factors are not mutually exclusive in explaining *SIS* changes. In order to investigate their influence, a study was carried out on the Mediterranean area as this region is one of the maritime areas with very high relative humidity and the largest aerosol loads in the world [1]. The authors used satellite data, specifically CERES (Clouds and the Earth's Radiant Energy System) SYN1deg products. Their results show that the spatial distribution of aerosol and water vapor are closely linked to the spatial distributions of its effects on solar radiation. The air constituents cause a *SIS* reduction between 2.0–8.0% due to aerosol and 11.5–15.0% due to water vapor effects. The authors found the greatest effects of aerosol and water vapor on *SIS* in the period from July to September.

Variations in *SIS* also cause variations in surface air temperature (*TAS*). Although *SIS* warms the air only during the day, its variability is related to the variability of daily *TAS* amplitude on monthly and decadal time scales [2].

Since data records are available up to 1970, a decrease in *TAS* due to the reduced input of *SIS* was detected [3]. The authors combined the temperature data with radiation data and additional components of the hemispheric energy budget. The relationship between the change in *SIS* and the change in *TAS* was applied to the reported trends in direct solar

radiation, revealing a probable cause–effect relationship. Long time series of temperature have already been tested several times. For Europe, the annual frequency of cold extremes has decreased since 1961 by about 7%/100 yr, with heat extremes increasing by more than 10%/100 yr [4]. Compared to mean annual warming of about 2–3 K/100 yr, the coldest winter temperatures have increased three times as fast, resulting in a smaller seasonal range and fewer variable winters. Changes in the warmest summer temperatures since 1961 show large spatial variations, with rates of change ranging from slightly negative to 6 K/100 yr. The spatiotemporal *TAS* change in Germany from 1951–2015, especially in spring, shows an increase in *TAS* of 1.6 K [5].

An earlier study carried out at the Baseline Surface Radiation Network (BSRN) station Tartu–Tõravere (58°16′ N, 26°28′ E) in Estonia found a good correlation between *SIS* and *TAS* [6]. The study also revealed the influence of the NAO index (*NI*) on European weather patterns and the *TAS* and *SIS* variability. The results of this work are discussed again later.

Since weather and associated *TAS* have an impact on the performance and electrical efficiency of photovoltaic (PV) systems, there is an interest in analyzing *TAS*-induced changes in PV system performance [7,8]. The PV system efficiency decreases with lower latitudes due to higher *TAS*. A study conducted in West Africa found that a *TAS* increase of 1.5 to 3.0 K decreases PV power generation by nearly 3.8% [9]. In mountainous regions, the performance of PV systems increases with decreasing *TAS* at higher elevations [10]. It was recommended to monitor *TAS* during PV system operation and to compensate for *TAS* increases accordingly to stabilize system performance and increase electrical efficiency [11].

Since 2000, the annual electricity generation from solar energy in Germany has increased from about 1% to more than 9% [12]. The aim of PV system operators is the increasing use of solar energy to mitigate anthropogenic climate change. During the transformation of the energy sector from conventional to renewable energies, it is imperative to maintain the balance between the variable energy supplies from solar and wind and the energy demand. The weather-related impacts of increasing penetration of the energy mix with solar and wind were exemplified in a previous study [13]. However, an increasing share of solar and wind energy in the future energy mix increases the likelihood of times when variable renewables are scarce. The greatest probability of occurrence of these periods, which in Europe are called "Dunkelflaute events", is winter [14]. This requires systems ensuring the base load of the energy supply.

Previous results [15] and newly available data stimulated the investigations reported in this paper. For the territory of Germany, we examined how variations in atmospheric circulation, expressed by the NAO index [16] and *SIS*, influence decadal *TAS* variability.

The abbreviations used in this manuscript are explained in Appendix A.

## 2. Materials and Methods

The analysis of the spatiotemporal *SIS* and *TAS* variability comprises the following steps: (1) Obtaining daily mean values of *SIS* and *TAS* with a horizontal resolution of $0.05° \times 0.05°$ for the 25-year period 1991–2015, (2) investigating the spatiotemporal variability of both variables by means of trend analysis and analysis of their inter- and intra-annual variability, (3) correlating *SIS* and *TAS*, and (4) correlating *SIS* and *TAS* with *NI*. The results were converted to a 10-year period (decade) to make them comparable with results from other studies.

All steps of data preparation and analysis were performed with the Satellite Application Facility on Climate Monitoring (CM SAF) R Toolbox software [17] and the Mathworks (Natick, MA, USA) Matlab software, version 2021b.

All figures showing the results for the study area are oriented north-south and have a scale of 1 cm equals 165 km.

### 2.1. Data

Daily mean values from the Surface Solar Radiation Data Set–Heliosat (SARAH)–Edition 2.1 were used for *SIS* analysis [18]. They are available on a regular grid. The *SIS* data records

are based on observations of the MVIRI and SEVIRI instruments on board of the first and second generations of METEOSAT satellites. Preparation and provision of *SIS* data were carried out by CM SAF [19]. Further details on the uncertainty of *SIS* are given in [20].

The daily mean *TAS* data records for the same period were taken from the HYRAS dataset [21], interpolated on the same grid as for *SIS,* and finally merged into one file containing all 25-year data records. This study [21] provides information on HYRAS data quality. Monthly *NI* means are available on the US National Weather Service's Climate Prediction Center website [16].

### 2.2. Assessment of Spatiotemporal Variability

The spatiotemporal variability and the long-term development of *SIS* and *TAS* were estimated by calculating mean values over the study period ($SIS_P$, $TAS_P$) at the grid cells and the study area. Monthly averages identified the *SIS* ($SIS_M$) and TAS ($TAS_M$) changes over the year. The long-term change (trend) in *SIS* ($SIS_T$) and *TAS* ($TAS_T$) is quantified by multiplying the slope of the trend lines by the respective interval length. The significance of $SIS_T$ and $TAS_T$ was determined by applying the Mann–Kendall test at a confidence level of $\alpha = 0.05$ [22].

In contrast to a previous study [23], the inter-annual variability (IAV) was not expressed in relative terms. To obtain absolute values that more clearly highlight the differences in magnitude between individual months and years, monthly and annual $SIS_{IAV}$ and $TAS_{IAV}$ values were calculated as the range (maximum minus minimum values) of differences in mean values.

### 2.3. Assessment of the Correlation between SIS and TAS

An immediate reaction of *TAS* to a change in *SIS* cannot be expected, as both natural and anthropogenic air pollution within the atmosphere lead initially to the absorption and scattering of *SIS*. A further reduction in *SIS* occurs due to the varying degree of coverage. [24]. Finally, *TAS* reacts to these influences. A study of the correlation between *SIS* and *TAS* can therefore only provide reasonable results for the four seasons. In addition, it should be considered that *SIS* values are available only for daylight hours, while *TAS* describes the course of air temperature over the whole day.

To find possible reasons for the relationship between *SIS* and *TAS*, data from *NI* were included in the correlation analysis. *NI* characterizes the atmospheric circulation between North America and Europe transporting air masses of different temperatures into the study area. *NI* is defined as the difference between the normalized winter air pressures at sea level in Lisbon and Stykkisholmur (Iceland) [25]. A positive *NI* is found if its contrast is higher than average due to a very low pressure over Iceland and a very high pressure over the Azores with strong westerly winds over the North Atlantic between 40° and 60° N. The corresponding circulation leads to mild winters and abundant rainfall over Europe to Siberia and the American East Coast, while over the Mediterranean to the Near East there is drought and relatively cold winters. With a negative *NI*, the pressure contrast between the Icelandic low and the Azores high is significantly reduced. This stands for weaker westerly winds and an occurrence of relatively mild and humid winters in the Mediterranean and cold and dry periods in Europe and on the American East Coast.

## 3. Results and Discussion

### 3.1. Annual Mean Pattern and Trends

Figure 1a shows the large-scale distribution of $SIS_P$ with decreasing values from south to north. The large-scale pattern is superimposed by regional and local patterns, which are mainly influenced by orography [20].

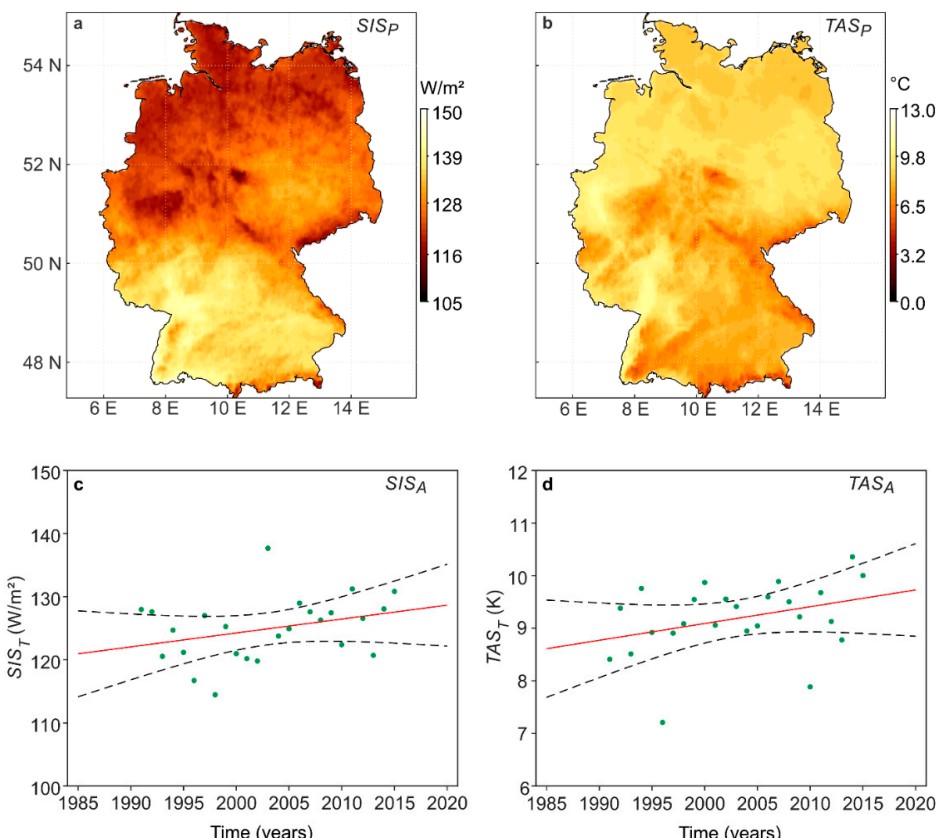

**Figure 1.** Mean values of (**a**) *SIS* (*SIS$_P$*), (**b**) *TAS* (*TAS$_P$*) at the grid cells in 1991–2015, trends of annual mean values of (**c**) solar radiation *SIS$_A$* (*SIS$_T$* = 2.4 W/m$^2$/10 yr, *R$^2$* = 0.13), and (**d**) surface air temperature *TAS$_A$* (*TAS$_T$* = 0.32 K/10 yr, *R$^2$* = 0.12) in Germany. The black dashed lines indicate the 95% confidence intervals for the regression models.

Figure 1b shows the *TAS$_P$* pattern in the study area. Remarkably high values occur in the Upper and Lower Rhine Area at more than 12 °C. These areas are particularly favored in summer by the inflow of warmer air masses from the southwest. Due to the more continental influence with low cloud cover especially in summer, the area of Eastern Germany shows slightly increased values of just about 10 °C. The urban heat islands of Hamburg, Berlin, Halle/Leipzig, and Munich are obvious. Furthermore, the elevations of the low mountains in Central and Southern Germany (Harz, Thuringian, and Bavarian Forest, as well as the Swabian Alb) are recognizable by low annual average values of about 4 °C. The long-term change in annual mean values of *SIS* (*SIS$_A$*) in Germany is positive but non-significant (Figure 1c). Their total change *SIS$_T$* is 2.4 W/m$^2$/10 yr. This result fits well with trends based on surface measurements in Central Europe reported in another study [26]. The relative standard deviation of *SIS$_A$* equals 3.9%, corresponding to an absolute value of 4.9 W/m$^2$. This matches with the findings of [27] who found a *SIS$_A$* variation with a relative standard deviation of 4–6% in most parts of Central Europe. A study of the spatiotemporal *SIS* variability over the Eastern Mediterranean in 1983–2013, using data records of *SIS* from the same source as used in this study, found a positive (brightening) and statistically significant *SIS$_T$* at the confidence level $\alpha$ = 95% with $2 \pm 0.05$ W/m$^2$/10 yr, with on- and offshore being almost equal [28].

The long-term change in *TAS* annual mean values (*TAS$_A$*) is significantly positive (Figure 1d) with a total change in *TAS$_T$* of 0.32 K/10 yr. A total change in *TAS* of 0.25 K/10 yr based on a 65-year data record was found in [5]. The relative standard deviation of the annual mean values of *TAS* equals 6.3%, corresponding to an absolute value of 0.57 °C.

The mean decadal changes of seasonal *SIS* and *TAS* and *SIS$_A$* and *TAS$_A$* values are summarized in Table 1. The annual and seasonal values for spring to autumn for $\Delta SIS$

represent more than 80% of all grid cells. Only 10% of the grid cells show a significant trend in winter. In contrast, the seasonal and annual trends of $\Delta TAS$ are only significant in autumn and to a lesser extent also for the year. Parallel to the positive trend of *TAS* in autumn, *SIS* also shows a similar trend. Only in winter does *TAS* not show any trend.

**Table 1.** Mean decadal changes of *SIS* ($\Delta SIS$) and *TAS* ($\Delta TAS$) in different intervals (year, season) in Germany 1991–2015. $R^2$ is the coefficient of determination.

| Interval | $\Delta SIS$ (W/m$^2$/10 yr) | $\Delta TAS$ (K/10 yr) |
|---|---|---|
| Year | +2.4, $R^2 = 0.13$ | +0.32, $R^2 = 0.12$ |
| Winter | +0.9, $R^2 = 0.08$ | −0.03, $R^2 = 0.00$ |
| Spring | +4.6, $R^2 = 0.08$ | +0.23, $R^2 = 0.03$ |
| Summer | +1.0, $R^2 = 0.01$ | +0.16, $R^2 = 0.02$ |
| Autumn | +3.1, $R^2 = 0.11$ | +0.66, $R^2 = 0.21$ |

In a study using the same *SIS* data records and the same period as the present study, but applied to the region of Piedmont (Northwest Italy), *SIS* showed an increase of about +2.5%/10 yr for annual averages, with the strongest trend in autumn (+4%/10 yr) [29]. Only in winter was a negative but non-significant trend found. A study of the spring period for the area of Germany using a 65-year *TAS* record (1951–2015) revealed an increase of 1.6 K for the whole period, or 0.30 K/10 yr [5]. This trend value is well in line with the corresponding spring value found in this study (Table 1).

Despite a noticeable increase in *SIS* in spring, this corresponds only with a small increase in *TAS*. Only with a delay of almost six months could a significant increase in *TAS* be observed in autumn. The reasons for this are: (1) the input of radiation energy into the atmosphere occurs only during daylight hours. Only in summer is the radiation input the largest due to the long daylight hours corresponding with the greatest elevation of the sun. The *TAS* values used in the study, on the other hand, cover the whole day. Therefore, the high amounts of solar energy available in summer can only lead to a significant increase in *TAS* in autumn. (2) Although the input of solar energy controls the different processes in the atmosphere and at the Earth's surface, it does not directly affect the change in *TAS*.

The inter-annual variability $SIS_{IAV}$ (Figure 2a) shows a greater variability of *SIS* than the long-term trend $SIS_T$. In the western parts of the study area, $SIS_{IAV}$ exceeds 45 W/m$^2$. $SIS_{IAV}$ decreases towards the east and northeast and fluctuates there in the range of 20–35 W/m$^2$. In the south, there is an area stretching west-east where $SIS_{IAV}$ is lower than in the surrounding areas. Here, in the Danube basin, $SIS_{IAV}$ is noticeably reduced compared to the surrounding areas. Near the northern coasts, the annual *SIS* variability is lowest, with $SIS_{IAV}$ values usually fluctuating in the range of 17–25 W/m$^2$. The mean and standard deviation of $SIS_{IAV}$ is $34 \pm 5$ W/m$^2$.

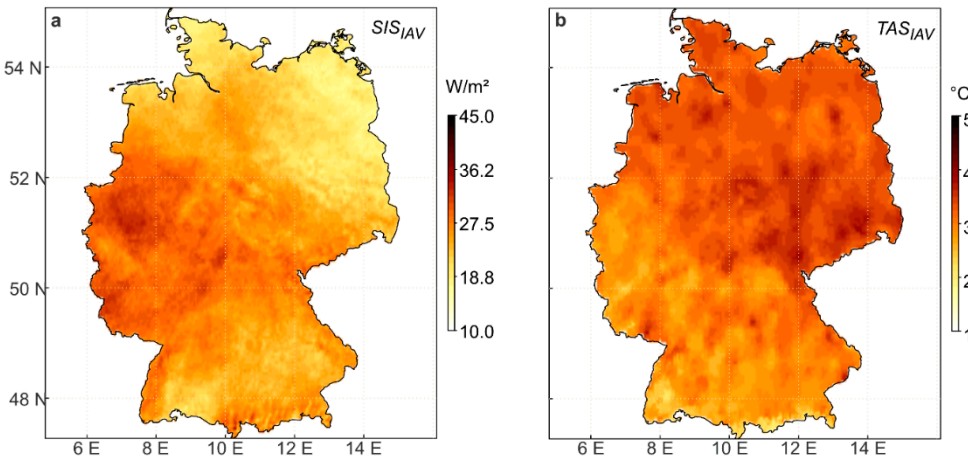

**Figure 2.** Inter-annual variability of annual means of (**a**) surface incoming solar radiation ($SIS_{IAV}$) and (**b**) air temperature ($TAS_{IAV}$) in Germany in 1991–2015.

The inter-annual variability of $TAS_{IAV}$ (Figure 2b) also shows a greater variability of $TAS$ than the long-term trend $TAS_T$. The mean spatial pattern of $TAS_{IAV}$ is roughly divided into two parts: In the south and southwest of Germany, $TAS_{IAV}$ is between 2 and 3 K, and in the north and northeast it is up to 5 K. The heat islands of the cities (Berlin, Halle/Leipzig, Dresden, Chemnitz, Hamburg, and Bremen) are clearly visible with values of 5 K. This confirms the findings of a previous study [5] mentioned above.

### 3.2. Trends and Inter-Annual Variability

Different methods of generating electricity require knowledge of the long-term trend of incoming solar energy. For this purpose, the long-term $SIS$ values ($SIS_T$) were calculated for each grid cell and month for the study period. As a result, Figure 3a shows the monthly frequency distributions of the trends in violin plots. The width of each individual violin is a measure of the frequency of the respective monthly trend. The upward and downward peaks of the violins indicate the extreme values of the monthly trends. The median values of the trends are given as numerical values at the respective black dots.

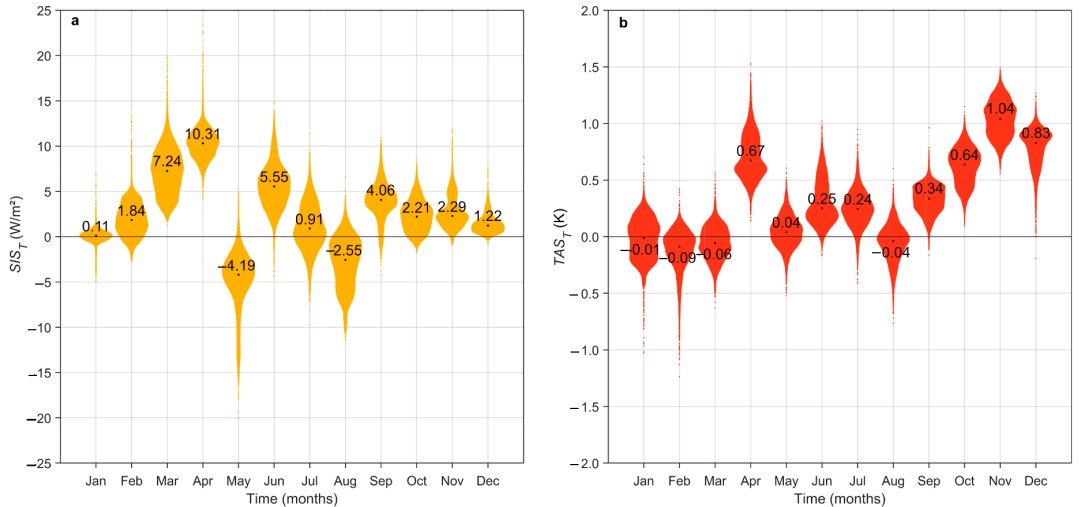

**Figure 3.** Violin plots of 10-year trend values of monthly (**a**) surface incoming solar radiation ($SIS_T$), and (**b**) air temperature ($TAS_T$) in Germany from 1991–2015 at the grid cells. The black dots mark the medians, which are given as numerical values.

The monthly $SIS_T$ values show a pronounced seasonal pattern. While local trends cluster around zero in January, they increase in February, March, and especially in April. The median $SIS_T$ values are highest in April at about 10 W/m²/10 yr. A significant decrease occurs in May. For this month, most $SIS_T$ values are negative, and the monthly median is $-4.19$ W/m²/10 yr, which is the lowest median of the whole year. In the second half of the year, the median monthly values of $SIS_T$ vary between $-2.55$ and $+5.55$ W/m²/10 yr. The scattering of the grid cell $SIS_T$ values is small and mainly limited to the range of $-10.0$ to $+10.0$ W/m²/10 yr.

The spatial distribution of the $SIS$ variability in the individual months shows a pronounced annual cycle. Here it should be mentioned briefly that $SIS_{IAV}$ shows a much wider range of variation in all months than $SIS_T$.

The violin plots for $TAS_T$ shown in Figure 3b were calculated using the same method as those violin plots of $SIS_T$ shown in Figure 3a. The long-term monthly grid cell values of the $TAS_T$ trend also show conspicuous values in two periods of the year (Figure 3b). April shows a noticeable long-term mean increase of $TAS_T = 0.8$ K/10 yr. From August to December, $TAS_T$ values increase continuously up to 1.0 K/10 yr. For April, another study found similar $TAS$ trend values of 0.5 K/10 yr for the north and 0.2 K/10 yr for the south of Germany [5]. The phenomenon of $TAS$ increase in late summer and early autumn has already been reported earlier [30].

The month with the most pronounced trends in *SIS* and *TAS* is April (Figure 4). The most significant increase of $SIS_T$ up to 15 W/m$^2$/10 yr can be seen in Northwestern Germany as well as in Bavaria. The negative trend of cloud fractional cover ($CFC_T$) in April of −4.1%/10 yr favors this development, further details on $CFC_T$ can be found in Figure 6 and are given in the corresponding text in [20]. From Eastern to Southwestern Germany, an area with a value of $SIS_T$ around 5 W/m$^2$/10 yr is discernible. Again, this behavior can be explained by the variability in *CFC*.

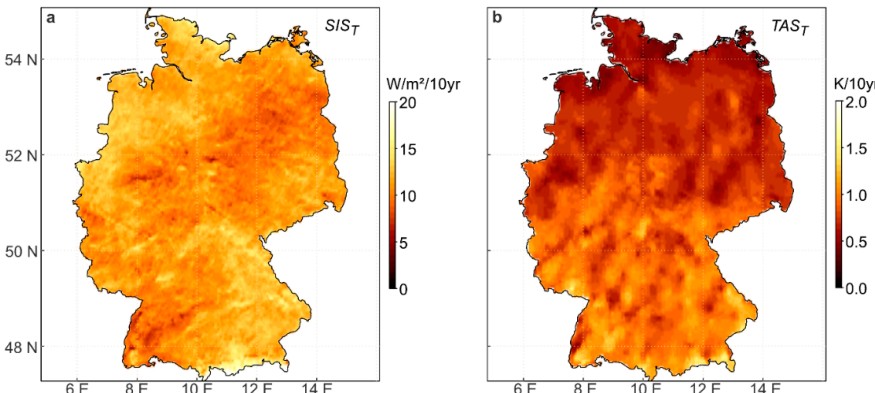

**Figure 4.** Ten-year trends of (**a**) *SIS* ($SIS_T$) and (**b**) *TAS* ($TAS_T$) in April in Germany from 1991–2015.

Regarding $TAS_T$ in April, Germany is divided into two parts: In the northern part there is hardly any trend (<0.5 K/10 yr), while the southern part of Germany shows values up to 1.2 K/10 yr. This is in line with a previous study [5] that found an average trend of *TAS* in Northern Germany of 0.30 K/10 yr and for Southern Germany of 0.28 K/10 yr for a 65-year period (1951–2015).

This underlines the well-known fact that there has been a significant increase in *TAS*, especially in the last decade of the study period analyzed by [5]. An increase in *TAS* can be expected when the *NI* is positive [31]. In almost all years, the *NI* values are positive in April.

Since Figure 3 shows the most striking trend in *SIS* and to some extent also in *TAS* in spring, the seasonal anomalies of *SIS* ($SIS_{MAM}$) and *TAS* ($TAS_{MAM}$) in spring are shown in Figure 5. Increasing $SIS_{MAM}$ coincides with an increase in $TAS_{MAM}$. In addition to the correlation between $SIS_{MAM}$ and $TAS_{MAM}$, the fluctuations of $SIS_{MAM}$ and $TAS_{MAM}$ within the study period were investigated. It was found that large fluctuations of $SIS_{MAM}$ are associated with large fluctuations of $TAS_{MAM}$. For Europe, a correlation coefficient between R = 0.73 and R = 0.86 was found in [2]. Especially for the years from 1990 onwards, the authors calculated high correlation values for the summer months.

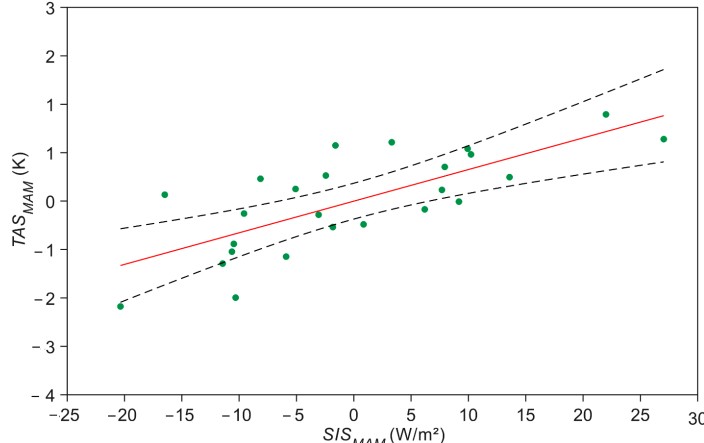

**Figure 5.** Anomalies of *TAS* ($TAS_{MAM}$) as a function of anomalies of *SIS* ($SIS_{MAM}$) in spring (MAM) in Germany from 1991–2015 with a root mean square error RMSE = 0.71 and $R^2$ = 0.55. The black dashed lines indicate the 95% confidence intervals for the regression model.

### 3.3. Changes in High $SIS_{MAM}$ and $TAS_{MAM}$ Values

For the dimensioning of PV plants, their operators do not need the long-term means of *SIS* and *TAS*, but rather, information on how often the given thresholds are exceeded. Corresponding results are given here.

Since the most significant changes in *SIS* and *TAS* are to be expected in spring, Figure 6a displays the spatial patterns of $SIS_{MAM}$. In Southern Germany, higher $SIS_{MAM}$ values are found than in the north. The low mountains in Central Germany are characterized by significantly lower $SIS_{MAM}$ values.

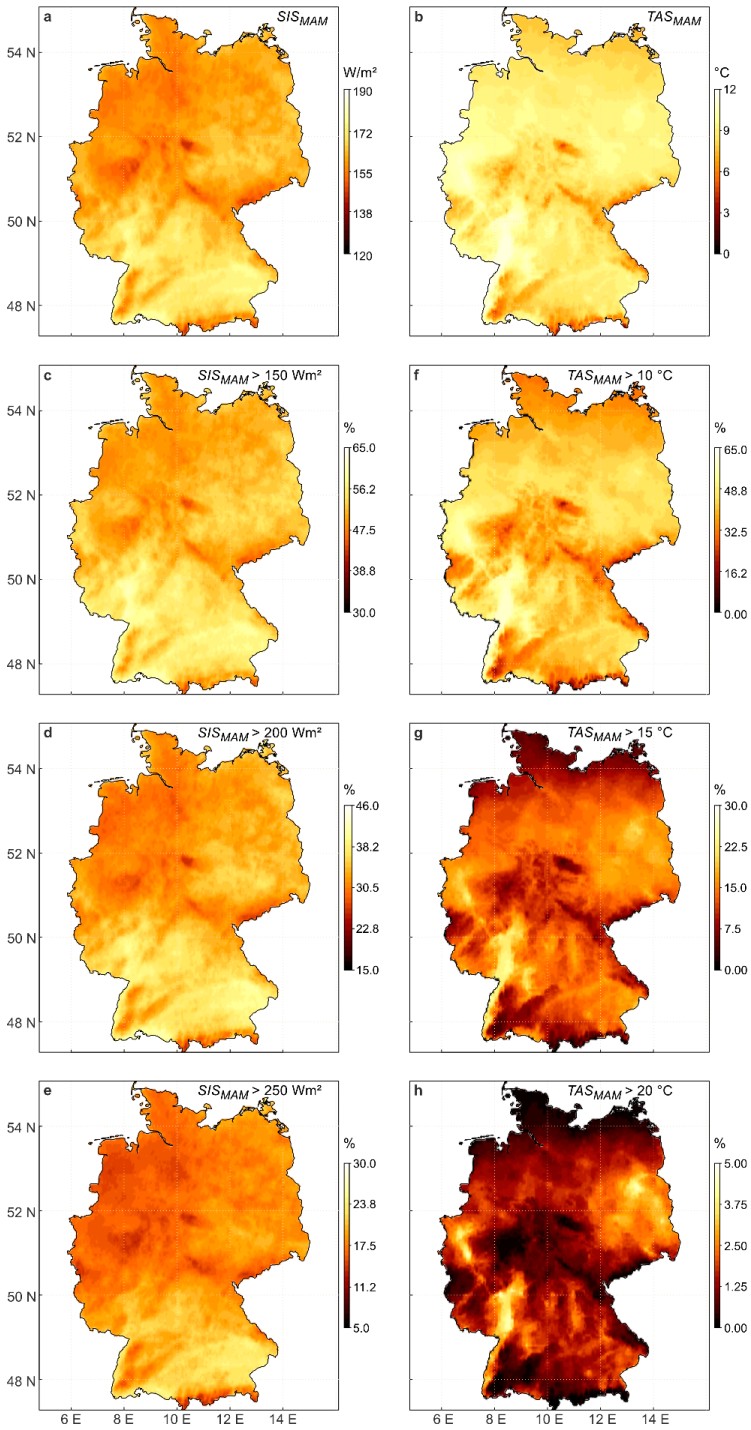

**Figure 6.** Seasonal means of (**a**) *SIS* ($SIS_{MAM}$) and (**b**) *TAS* ($TAS_{MAM}$) in spring (MAM). Share of events where $SIS_{MAM}$ or $TAS_{MAM}$ at each grid point is greater than (**c**) 150, (**d**) 200, (**e**) 250 W/m$^2$, and (**f**) 10, (**g**) 15, (**h**) 20 °C in Germany from 1991–2015.

The long-term means of $SIS_{MAM}$ are between 120 and 190 W/m². In cases of cloudless sky, the average daily irradiance in the study area may be up to 500 W/m².

Since hardly a significant number of events could be detected above 300 W/m², the number of $SIS_{MAM}$ values exceeding the thresholds of 150, 200, and 250 W/m² was investigated. Figure 6c–e show the percentage of all cases of $SIS_{MAM}$ exceeding the thresholds set in the spring months. A limit value of $SIS_{MAM}$ = 150 W/m² is exceeded in at least 50% of all grid cells in almost the whole of Germany. Raising the threshold to $SIS_{MAM}$ = 200 W/m² and $SIS_{MAM}$ = 250 W/m² limits high shares of events to the southern parts of the country.

High $TAS_{MAM}$ > 10 °C mainly occur in the southwestern and western parts of the study area (Figure 6b). Lower $TAS_{MAM}$ values of about 5 °C can be observed in the low mountains and the foothills of the Alps. The average mean values of $TAS_{MAM}$ are between 0 and 12 °C. In the study period the average maximum $TAS_{MAM}$ = 27 °C. Figure 6f–h show the percentage of all $TAS_{MAM}$ cases exceeding the thresholds of 10, 15, and 20 °C in the spring months.

In at least 50% of all cells, $TAS_{MAM}$ = 10 °C is exceeded. The lowlands are marked with 15% and the Rhine Valley with 60% of all the cases. The limit value of $TAS_{MAM}$ = 15 °C is exceeded in 22% of all cases in the Rhine Valley. In the other areas of Germany, this limit is exceeded in between 5% and 10% of all cases. The limit value of 20 °C is exceeded in only 3% of all cases.

Additional information on the long-term increase of $SIS_{MAM}$ and $TAS_{MAM}$ can be obtained by studying the increase in the number of events in individual $SIS_{MAM}$ and $TAS_{MAM}$ ranges. Due to the development of solar energy applications over recent years, there is an increasing need to study the transition of $SIS_{MAM}$ and $TAS_{MAM}$ to higher levels. Therefore, the long-term change in the number of events in the $SIS_{MAM}$ range of 150–200 W/m² and in the $TAS_{MAM}$ range of 10–15 °C is displayed in Figure 7. The trends of $SIS_{MAM}$ and $TAS_{MAM}$ significantly increase.

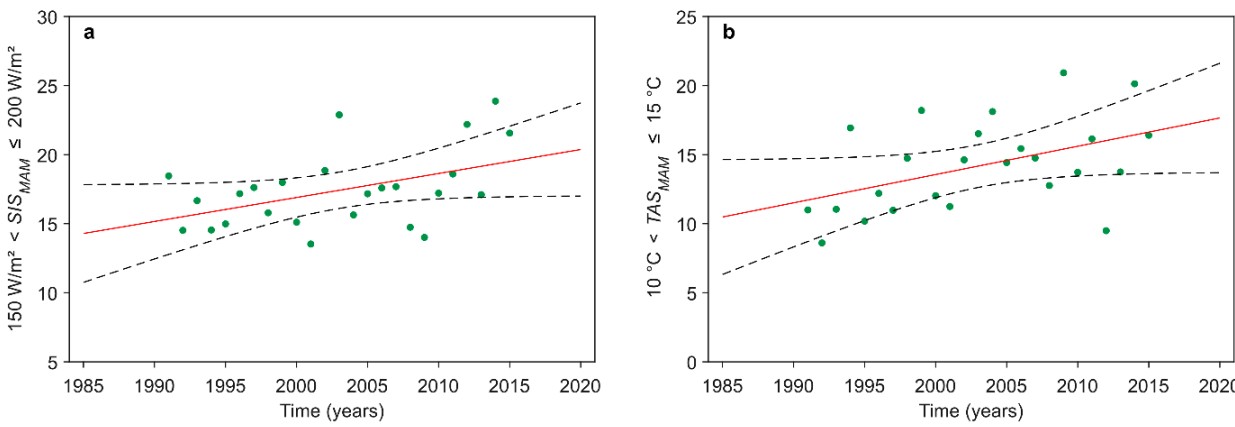

**Figure 7.** Long-term change in (**a**) solar radiation ($SIS_{MAM}$) in the range 150 W/m² < $SIS_{MAM}$ ≤ 200 W/m² (1.73 W/m²/10 yr and $R^2$ = 0.22) and (**b**) surface air temperature ($TAS_{MAM}$) in the range 10 °C < $TAS_{MAM}$ ≤ 15 °C (2.1 °C/10 yr and $R^2$ = 0.21) in spring (MAM) in Germany from 1991–2015. The black dashed lines indicate the 95% confidence intervals for the regression model.

*3.4. Influence of the NAO Index on SIS and TAS*

As an example, for April, the correlations between *NI* and *SIS* ($SIS_{April}$) and *NI* and *TAS* ($TAS_{April}$) are shown in Figure 8.

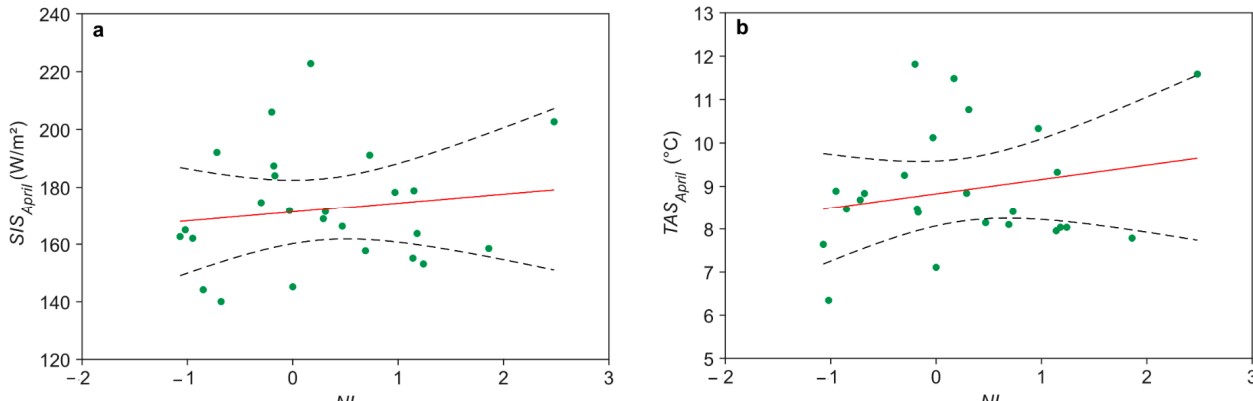

**Figure 8.** Change in (**a**) $SIS_{April}$ ($\Delta NI/\Delta SIS_{April}$ = 3.16, $R^2_{NI,SIS}$ = 0.021) and (**b**) $TAS_{April}$ ($\Delta NI/\Delta TAS_{April}$ = 0.33, $R^2_{NI,TAS}$ = 0.05) in dependence on the NAO index (*NI*) in April in Germany 1991–2015. The black dashed lines indicate the 95% confidence intervals for the regression model.

Both figures show a weak dependence of $SIS_{April}$ and $TAS_{April}$ on *NI*. Similar correlations for all months are given in Table 2.

**Table 2.** Correlation coefficients (*R*) between monthly values of *SIS* and *TAS* ($R_{SIS,TAS}$), *NI* and *SIS* ($R_{NI,SIS}$), and *NI* and *TAS* ($R_{NI,TAS}$) in Germany from 1991–2015. *R* values on significance level $p < 0.05$ are in bold.

| Month | $R_{SIS,TAS}$ | $R_{NI,SIS}$ | $R_{NI,TAS}$ |
|---|---|---|---|
| January | −0.15 | 0.14 | **0.50** |
| February | 0.21 | **0.40** | **0.43** |
| March | 0.33 | **0.35** | **0.66** |
| April | **0.68** | 0.15 | 0.22 |
| May | **0.84** | 0.31 | 0.34 |
| June | **0.74** | 0.32 | 0.00 |
| July | **0.93** | 0.38 | 0.32 |
| August | **0.76** | 0.43 | 0.29 |
| September | **0.74** | 0.31 | 0.21 |
| October | 0.30 | −0.06 | 0.15 |
| November | 0.43 | **0.35** | −0.04 |
| December | 0.43 | 0.21 | **0.80** |

We found, similar to the study of [6], an interdependence of *SIS*, *TAS*, and *NI* of about 0.40–0.60 for the study area.

With the exception of January, the monthly $R_{SIS,TAS}$ is positive. In July, which has the longest light days, $R_{SIS,TAS}$ = 0.93. The correlation from April to September is significant at $p < 0.05$.

In the winter months, the high and also significant $R_{NI,TAS}$ values are striking. During this time, *TAS* is essentially determined by high *NI*, which is responsible for a pronounced westerly wind circulation. The same was found for the BSRN station Tartu–Tõravere in Estonia [6] which indicates that atmospheric circulation is an important factor controlling *TAS* in the study areas.

The results in [6] differ slightly from those of our study for several reasons: (1) we used data records from a larger area (48° N–55° N/6° E–15° E) rather than from a single station (BSRN station Tartu–Tõravere in Estonia), (2) the station is located about 5° further north (about 500 km) than the northernmost point of Germany. Thus, in the winter months, the days of light are significantly shorter than those in Germany. In the summer months, the ratio reverses in favor of Tartu-Tõravere. (3) The two study periods are not identical: The author [6] used data records from the period 1955–2007.

### 3.5. Correlation between SIS and TAS

The similar long-term increase in *SIS* and *TAS* in April and October presented in Figure 3 stimulated a correlation analysis of these two variables. A good correlation between these variables was also found by [3]. Figure 9a–d show the corresponding results for the four seasons. Only in spring and autumn, is a clear dependence of *TAS* on *SIS* discernible. The similar increase in these two variables in these two seasons indicates their good correlation.

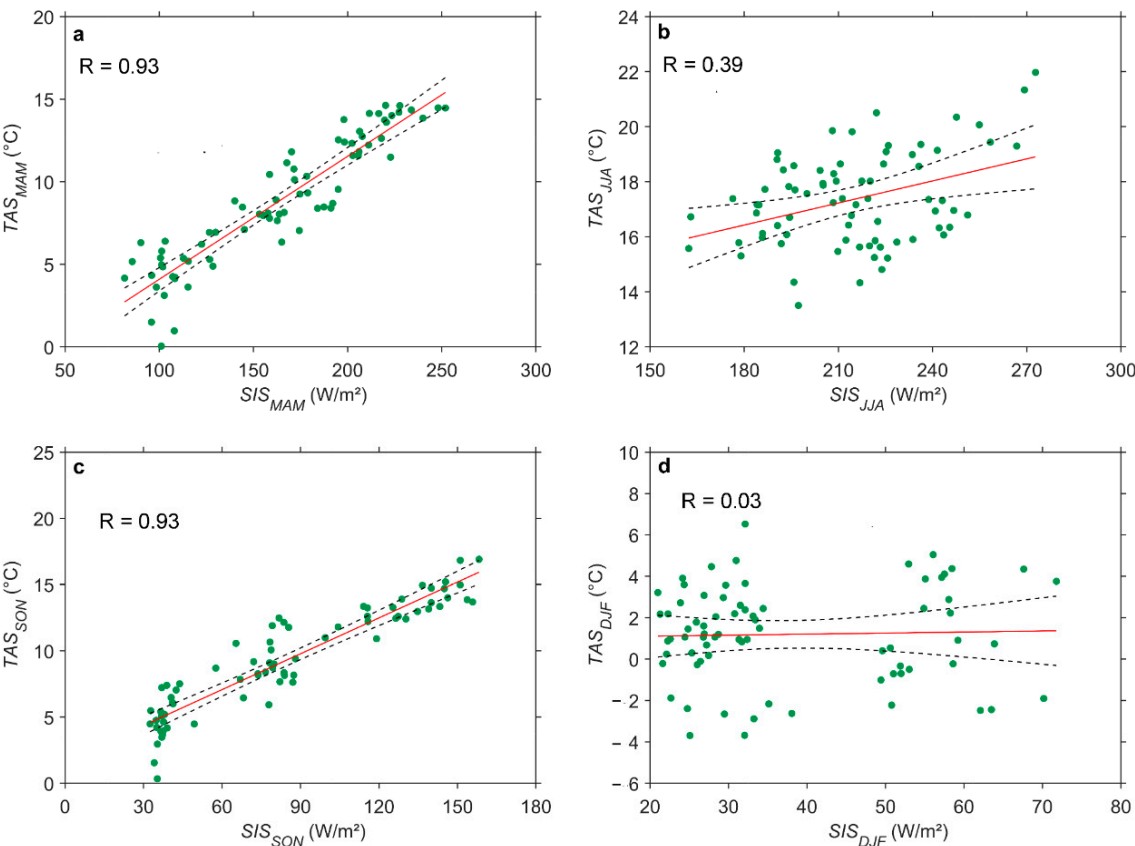

**Figure 9.** Correlation of seasonal means of *SIS* and *TAS* in (**a**) spring (MAM), (**b**) summer (JJA), (**c**) autumn (SON), and (**d**) winter (DJF) in Germany from 1991–2015. The correlation coefficient (*R*) is given for each season in the corresponding figure. The black dashed lines indicate the 95% confidence intervals for the regression model.

### 3.6. Ratio of Temperature and Solar Radiation

Finally, the regional structure of the long-term ratio TAS/*SIS* in the four seasons is given in Figure 10a–d. In all four seasons, the ratio is between 0 and 0.15 K/W/m$^2$. The larger the quotient (light areas), the more *TAS* reacts on an increase in *SIS*. This is clearly visible in all four seasons in the large basins in Southwestern Germany. While spring and autumn have hardly any regional structure but a strong correlation, the strong increase in autumn is particularly noticeable in Northern Germany. In winter, the sign of K/W/m$^2$ reverses in Southern Germany and in the low mountains, while in the north it remains positive. This regional two-part behavior is also evident from the fact that the correlation between *SIS* and *TAS* is unstructured in these seasons. This behavior is determined on the one hand by changes in the *NI* and the lower inter-annual variability of cloud cover [20].

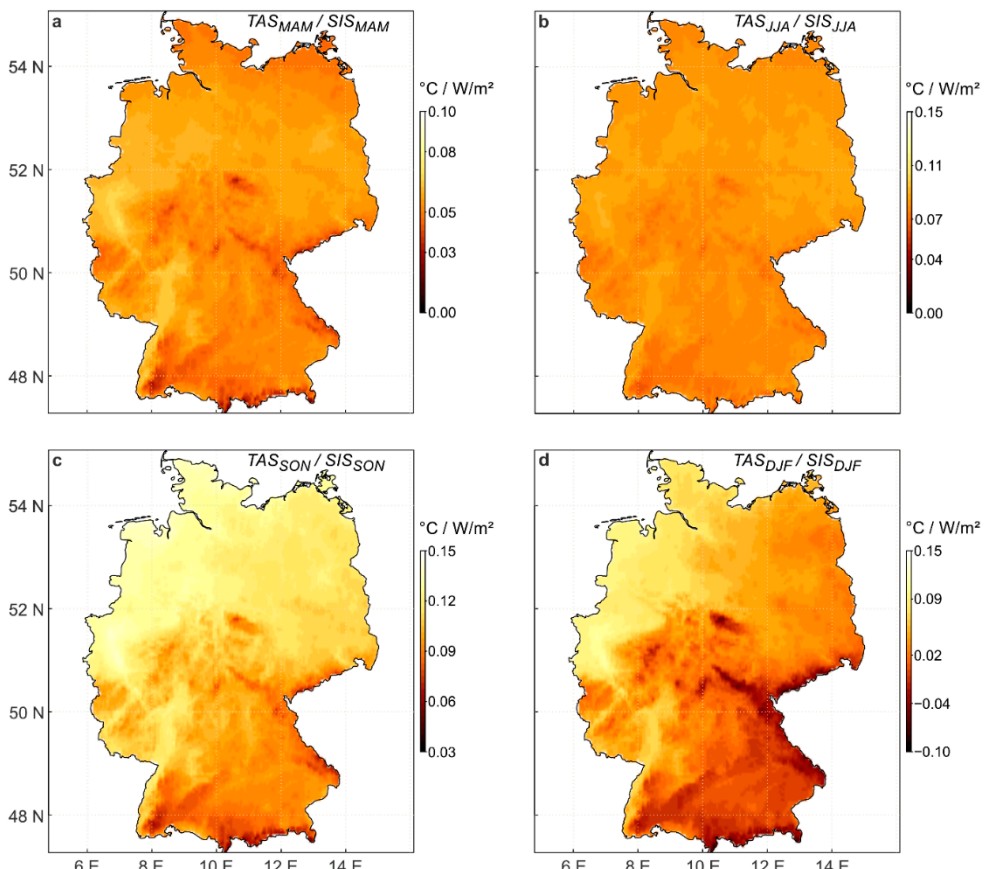

**Figure 10.** Ratio of surface air temperature (*TAS*) and solar radiation (*SIS*) in (**a**) spring (MAM), (**b**) summer (JJA), (**c**) autumn (SON), and (**d**) winter (DJF) in Germany 1991–2015.

## 4. Conclusions

To mitigate climate change, solar energy must play an increasingly important role in today and tomorrow's global energy supply. Before deciding to install PV systems in a particular area, a thorough analysis of the long-term spatial and temporal trends of solar radiation and surface air temperature is required. The following characteristic parameters of surface incoming solar radiation and surface air temperature are important for PV system operators to optimize their systems: long-term averages in the individual regions, the long-term trend of mean values and their variability, and finally an increase in exceedances of specified thresholds.

For the climate of the study area, we found a simultaneous long-term increase in incoming solar radiation and air temperature in spring. A further increase in surface air temperature was observed in autumn. The annual variability of incoming solar radiation and surface air temperature is much greater than their long-term variability.

Although an increase in solar radiation does not directly result in an increase in surface air temperature, an influence of solar radiation on surface air temperature was shown.

It was found that the *TAS*/*SIS* ratio is always positive, except in winter. This indicates a slight but almost equal interdependence of the two variables in the whole study area. Due to the short period of light days in winter, the ratios are unclear in this season.

The variable input of solar radiation in the study area influences the large-scale circulation, expressed by the variability of the NAO index. This quantity then controls the change in surface air temperature. These interdependencies are responsible for the efficiency of the PV systems and must therefore be considered by the operators of PV systems.

**Funding:** This research received no external funding.

**Data Availability Statement:** SIS-data are available online: https://wui.cmsaf.eu/safira/action/viewProduktDetails;jsessionid=A57DBBCF8B91A1018C4978343231BB1B.ku_1?fid=28&eid=21833_22009, TAS-data are available online: https://opendata.dwd.de/climate_environment/CDC/grids_germany/daily/hyras_de/air_temperature_mean/, NAO-Index-Data are available online: https://www.cpc.ncep.noaa.gov/products/precip/CWlink/pna/norm.nao.monthly.b5001.current.ascii.

**Conflicts of Interest:** The author declares no conflict of interest.

## Appendix A

| Acronyms, Abbreviations | |
|---|---|
| CM SAF | Satellite Application Facility on Climate Monitoring |
| HYRAS | Hydrometeorological Gridded Data from DWD |
| NOAA | National Atmospheric and Oceanic Administration |
| SARAH | Surface Solar Radiation Data Set-Heliosat |
| **Symbols** | |
| $\alpha$ | confidence level |
| $CFC$ | cloud fractional coverage (%) |
| $IAV$ | inter-annual variability |
| NAO | North Atlantic Oscillation |
| $NI$ | NAO index |
| $p$ | significance level |
| $R$ | correlation coefficient |
| $R^2$ | coefficient of determination |
| $SIS$ | surface incoming solar radiation ($W/m^2$) |
| $TAS$ | air temperature standard measured 2 m above ground ($^{\circ}$C) |
| **Subscripts** | |
| A | year |
| DJF | winter |
| IAV | inter-annual variability |
| JJA | summer |
| M | month |
| MAM | spring |
| NAO | North Atlantic Oscillation |
| P | study period (1991–2015) |
| SIS | solar radiation |
| SON | autumn |
| T | trend |
| TAS | surface air temperature |

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
