# Peer review of "Trends and Interdependence of Solar Radiation and Air Temperature—A Case Study from Germany"

_2674-0494, doi:10.3390/meteorology1040022_

Round 1

Reviewer 1 Report

This article presents an interesting study on the spatiotemporal variation of solar irradiation and ambient air temperature in Germany, using a long-term statistic of realistic modeled data. The analysis is informative, scientifically sound and contains interesting information. I only have some minor observations and I recommend its publication.

* I think the article should precise its definition of variability. The term variability can represent different things. For instance, in the field of solar energy and solar radiation, there are different definitions and metrics for its quantification, depending on the time scale. As far as I can see, this article uses the term variability for two things: the interannual variability, and the change in the long-term values per year. I suggest differentiating these two terminologies. From my perspective, variability is a proper term for the former, but for the latter I would rather use long-term trend or long-term change (increase/decrease) per year. Authors can choose another terminology, of course, but my suggestion here is to clearly differentiate these concepts.

* Following the previous bullet, Figure 3 requires a clear explanation on what it is being shown. In particular, it should be clear that this plot is not showing the inter-monthly variability (defined as in the sense of the interannual variability).

* I think the article should better highlight its novelty and contribution in the frame of pre-existing climatological studies about solar radiation and air temperature in Germany. A literature revision on this is also required, to put context on the contributions of the present work.

* The data uncertainty is not addressed. I suggest including in section 2.1 the expected data uncertainty, based on previous works evaluating the SARAH and HYRAS data sets against ground measurements with controlled quality.

* Lines 267-271 provide an analysis for the Spring season. I wonder why the other seasons are not also addressed. I think there should be a word on this.

* Lines 288-292: it would be very useful to have a clear sky day reference here. Which is the average daily irradiance expected in a clear sky day for each season? I would help to put these thresholds in context.

* The argument for choosing Germany as the target region, that is included in the abstract and in the introduction, is weak and not needed, in my opinion. I think it is fair for the authors to choose his country as the target for this work, so no further explanations are required. I suggest removing it, as a soft suggestion, of course.

* In the introduction it should be clearly explained that SIS refers to global solar downward radiation at a horizontal plane (I suppose, as it is the most common solar radiation magnitude).

MANUSCRIPT DETAILS:

* Lines 36-38: In which time scale and condition? Annual? Clear sky? All-sky resource? Aerosols’ impact is mostly noticeable in clear sky irradiance.

* Line 48: “... is about 0.40-0.60 for that location”.

* Line 52: “mandatory and essential” -> I do not agree with this. Daily cleaning is mandatory only in some climates, like desertic ones, in which aerosols and sand winds are deposited over the panels. In most climates monthly cleaning is just recommended and in some others a seasonal cleaning is more than adequate.

* Line 61: “lower” -> “higher”, isn’t it?

* Lines 90-91-> “the use of solar energy, to maintain a balance between the variable supply of renewable energy and the variable demand for energy”. I did not understand this, solar energy is not controllable, thus the argument is actually the opposite. Maybe there is some misspelling here.

* Line 256: CFC is not defined here, only in the Appendix.

Author Response

Find attached my comments to reviewer 1.

Reviewer 2 Report

Dear Authors,

After I spent a while read the manuscript entitled “Trends and Interdependence of Solar Radiation and Air Temperature—A Case Study from Germany”, I found the manuscript presented useful results for wide areas. However, I have few comments to please consider for improve the manuscript, as the list below:

- This manuscript should has a flowchart figure showing the logics and conditions for the data processing. 

- Line 29: should give the full phase of the abbreviation before using (SIS), and please recheck other abbreviation.

- Maps in Figure 1, 2, 4, 6 and 10 should use Red-Green-Blue color scale instead of dark-light orange color scale.

- Every maps should has a compass and a length scale.

Best regards

Author Response

Find attached my comments to reviewer 2.
